# Steering Language Models with Game-Theoretic Solvers

## Abstract

Mathematical models of strategic interactions among rational agents have long been studied in game theory. However the interactions studied are often over a small set of discrete actions which is very different from how humans communicate in natural language. To bridge this gap, we introduce a framework that allows equilibrium solvers to work over the space of natural language dialogue generated by large language models (LLMs). Specifically, by modelling a dialogue task in terms of the players, strategies and payoffs of the "game" of dialogue, we can create a binding from natural language interactions to the conventional symbolic logic of game theory. Given this binding, we can ask existing game-theoretic algorithms to provide us with strategic solutions (e.g., what string an LLM should generate to maximize payoff at equilibrium), giving us predictors of stable, rational conversational strategies that current LLMs can employ when generating dialogue. We focus on three domains that require different negotiation strategies: scheduling meetings, trading fruit and debate, and evaluate a state-of-the-art pre-trained LLM's ability to generate language when guided by solvers. Our evaluation assesses whether LLMs are more strategic against their partners when guided by equilibrium solvers and whether the language generated under these solutions results in higher payoff. We see that LLMs that do follow game-theory solvers result in dialogue generations that are less exploitable than the control (no guidance from solvers) in our three negotiation domains. We discuss future implications of this work, and how game-theoretic solvers that can leverage the expressivity of natural language can open up a new avenue of guiding language research.

## 1 Introduction

Existing large language models (LLMs) have achieved remarkable performance on many natural language tasks (Radford, 2020; Jiang et al., 2023; Team et al., 2023). They can generate fluent and coherent text, answer questions and serve as tools for many downstream applications that are helpful to human users (Schick et al., 2024). However, there are many studies showcasing the failures in the reasoning capabilities of such models. For example, even when the surface form of LLM-generated text appears plausible and human-like, they often fail to exhibit rational and consistent reasoning strategies (Jiang et al., 2020; Turpin et al., 2024). In a different area of research however, the study of rational and strategic behaviour amongst agents has long been studied in the field of game theory (Fudenberg & Tirole, 1991; Rosenschein & Zlotkin, 1994), where mathematical models and algorithms can solve for such optimal behaviour allowing agents to win complex games. These models have applications in a wide range of fields, from social science, economics and cognitive science (Stone & Veloso, 2000), allowing us to use this formulation to build agents exhibiting intelligent reasoning strategies against competitors or cooperators in multi-agent settings. For example, in games like Diplomacy and StarCraft (Vinyals et al., 2019; FAIR et al., 2022), methods that can solve for an equilibrium state of a game have allowed agents to exhibit strategies that get them to human-level performance in games that require complex decision making.

However, most of these success stories are on games in the colloquial sense of the word. These are card, board, or video games that often have action spaces far smaller than the space of natural language words. Language also comes with the underlying complexities of strategizing about beliefs and intents of players over the semantics of the generated words. In this work we ask whether we can combine outputs from game-theoretic models (e.g., ones that can solve for optimal equilibrium solutions) to guide language model generations in strategic games. We show how we do not have to be restricted to traditional "games" studied in game theory, but can extend to standard dialogue tasks, where two LLMs generate natural language to communicate with each other. We do this by creating a mapping from natural language dialogue tasks to the formal framing of imperfect information games, thus allowing off-the-shelf equilibrium solvers to find the optimal action an agent should be taking. When information from such a solver is then fed back into an LLM, we show how this can lead to more rational and strategic natural language generations from LLMs in dialogue games. In three different domains, we evaluate the extent to which a game-theory guided LLM outperforms a standard LLM and also analyze the effect of different types of equilibrium solvers on LLM generations.

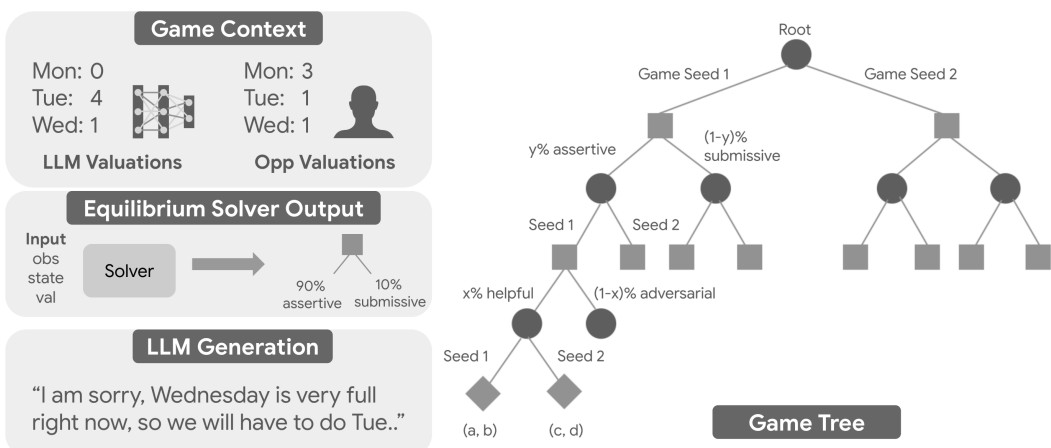

Figure 1: Figure shows an overview of our framework. On the left we see an example dialogue game in a meeting scheduling domain. On the right is a dialogue game tree showing equilibrium solver decisions. The squares denote decision points at which a solver chooses between actions. Values below the diamonds correspond to the payoffs for player 1 and player 2, respectively.

In summary, our contributions are as follows. We first develop a binding from conversational dialogue to the language of game theory and define the relevant components needed by any equilibrium solver to be able to solve the task. Given this framing, we use off-the-shelf game-theory models to solve for equilibria. The equilibrium strategy is then passed back to an LLM as part of its context, that it can use to guide the next natural language response it generates to its opponent. We then evaluate whether the influence of the equilibrium solvers does indeed lead the LLM to generate more strategic responses and "win" the game against their opponent. The remainder of this paper is framed as follows. In Section 2 we introduce our framing of dialogue as a formal game by mapping to key concepts of game theoretic models, and discuss solution concepts to find equilibria for games. We show how these solvers when combined with LLMs can work in three dialogue domains (Section 3) and present experiments providing empirical support for using game-theoretic models to improve LLM outputs (Section 4). Lastly, we outline related work in the area, and discuss the limitations and also implications of this work.

## 2   Framing Dialogue as an Extensive-Form Game

Extensive-form imperfect information games represent one of the most general classes of games in game theory. They allow us to represent sequential (i.e., temporally-extended)

interactions between players and their underlying valuations or preferences that influence the actions they take in a game. In this section we outline our formal framing of dialogue as a traditional extensive-form game i.e., a tuple $\langle \mathcal{N}, c, \mathcal{A}, \mathcal{H}, \mathcal{Z}, u, \tau, \mathcal{S} \rangle$ and define all the necessary elements of a traditional game below.[1]

- $\mathcal{N} = \{1, 2, \cdots, n\}$ is a set of $n$ **players**. There is also a special player, $c$, called **chance** (or "nature").

- $\mathcal{A}$ is a finite combinatorial set of **actions** that can be taken. In the case of dialogue, LLMs get a game context in the form of a string as input and the action space can be the strategy to be taken (e.g., to be assertive). Note, one could extend the action set to include a choice of which LLM (out of a set of models of different sizes) to call as well; the only requirement is that this space remain finite.

- $\mathcal{Z} \subseteq \mathcal{H}$ is a set of **terminal histories** determined by either a limit on the allowable tree-depth (we only allow a finite number of messages per player) or by an LLM tasked with determining whether a conversation has ended e.g., a deal has been struck and no further conversation is required.

- $u : \mathcal{Z} \to \Delta_u^n \subseteq \Re^n$, where $\Delta_u = [u_{\min}, u_{\max}]$ is a utility (or payoff) function assigning each player a payoff at the end of the game. We construct a prompt specific to the domain at hand and let an LLM quantify the reward to each player following previous work (Kwon et al., 2023; Wei et al., 2022).

- $\tau : \mathcal{H} \to \mathcal{N} \cup \{c\}$ is a player identity function; $\tau(h)$ indicates whose turn it is.

- $\mathcal{S}$ is a set of **infostates**. Each infostate $s$ represents a partition of $\mathcal{H}$ such that each $h \in s$ cannot be distinguished by players other than $\tau(s) = \tau(h)$ for all $h \in s$. In our games, we implicitly define these partitions by defining what information is public versus private to player $\tau(s)$, and a player's infostate therefore includes observable dialogue history, action history and private information. In each of our games, all sent messages are appended to a public thread making the entire dialogue history public knowledge. Note this still omits certain private information like each player's action, e.g., the precise *tone* a message is written in although this might be approximately inferred by a player. We only require each player have knowledge of their own action history to maintain **perfect recall**[2].

**Solution to Games** Given this definition of a game, we can define a metric which describes desirable outcomes of the players of the game. This usually takes the form of players maximizing their reward (also called payoffs) or minimizing some cost. Note that this is an inherently multi-objective problem: in games, the behaviour of players influences the payoffs of the other players. The interactions between agents may be purely competitive, purely cooperative, or a mixture between the two. Multi-objective optimization results in a Pareto-front of possible solutions that often also include a notion of stability (or *equilibrium*): no player should have incentive to unilaterally deviate away from that solution. Therefore, each game can have a set of combinations of player actions that are in equilibrium.

As an example, the most famous solution to a two-player zero-sum game is John von Neumann's minimax solution (von Neumann, 1928), that guides a player to choose each next move by computing the value of each state of the game, and then guiding players to take actions that result in a minimum value of the opponents state. At this equilibrium, the resulting actions are unexploitable, and if there are multiple solutions, they are interchangeable. The Nash equilibrium (Nash, 1951) is a famous generalisation of this solution concept to many-player games—it provides a strategy profile (meaning a mixed strategy for every player in the game) under which no player has an incentive to unilaterally deviate. Many other rich solution concepts have been defined (Aumann, 1974; Hannan, 1957)[3] and we use similar solutions that we refer to as **equilibrium solvers** to solve the games we define.

---

[1]Note that the exact binding is not unique and many possible ways of mapping a new game to this formalism are possible. We will explore an approach later that can help to modify and improve this model automatically given an initial model.

[2]Without this property, solving a game can be substantially more expensive computationally.

[3]See Appendix I for tractability of solution concepts.

**Our framework**   Given this formulation above we can frame a dialogue task between two agents as a two-player extensive-form game. The players are LLMs that can generate natural language responses, the actions that an equilibrium solver guides an LLM towards can be "tones" or strategies the model should exhibit while producing a response, and player valuations are defined in the input context given to the model. Separate LLMs can act as judges or reward models to assign payoffs to the agents at the end of the game by evaluating the responses generated. We outline details of this methodology in the next section.

## 3   Experimental Methodology

### 3.1   Models

We use two autoregressive language models from the PaLM model family (Anil et al., 2023). The models are pretrained on on the PaLM-2 pretraining corpus and then used at inference-time to generate dialogue utterances. Specifically, for a given domain, a model $P$ conditions on a prompt that contains information about the domain, and a few turns of dialogue that it is then required to respond to. When using input from the game-theoretic solvers, the prompt includes this information that therefore influences the LLM generations. We provide example prompts in in Appendix C. We also use LLMs as reward models $R_{LM}$, where the LLMs are given few-shot chain-of-thought-prompting samples to allow the model to calculate trade values as shown in Appendix D.

### 3.2   Dialogue Game Domains

We consider three different dialogue domains shown in Figure 2 that require strategic reasoning between two agents. These games are procedurally generated by defining elements in the game (e.g., items to trade) or characteristics to exhibit (e.g., different tones) that can form the initial context for the dialogue. We generate 1000 such games for each domain by iterating over sets of items and action spaces, and use these to evaluate our framework. The LLM gets prior information about the game domain in it's input context as part of it's prompt and is required to generate a response. When it is given guidance and additional input from a game theory solver (e.g., to tell it what tone to take) we aim to evaluate if this solver input modulates it's response such that the augmented generation is significantly different or more strategic than the baseline LLM response. We describe the domains below.

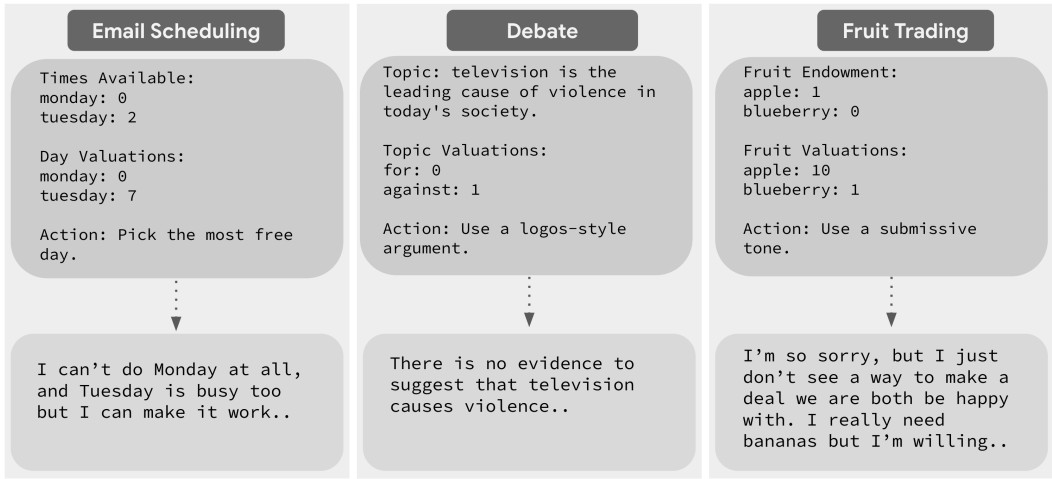

Figure 2: Figure shows the three dialogue domains we consider: an email scheduling task on the left, a debate task in the centre, and a fruit trading task on the right.

**Scheduling Meetings**   In this domain players are required to schedule a meeting in a multi-turn negotiation setting. Each player begins with a set of allowable days of the week they

are available to meet in (e.g., "monday" and "tuesday") and have non-negative valuations over each day of the week. Both of these pieces of information are private to the players. Players can choose to reveal this information in their natural language generations if they wish and their actions here are the days of the week on which they propose to meet. The game rewards players according to how much they value the agreed upon day, and both players receive zero reward if no agreement is made. Here, the equilibrium solvers can guide LLMs on which action (i.e., the set of days of the week) might be the most optimal given the context history of the game and the players valuations.

**Trading Fruit** We use a fruit trading domain similar to (Chawla et al., 2023), where each player begins with a private endowment of types of fruit as well as private valuations over the items. Players are rewarded by the difference in value between their basket after the trade and that before the trade. Previous work on this game theoretic task has argued that personality or tone has an impact on negotiations in natural language and we use the equilibrium solver guidance to modulate the tone of the LLM generations (e.g., submissive, assertive and so on). There are four possible tones to choose from and we outline the actions and example prompts from this game in Appendix D.

**Public Debate** We also consider the domain of debate (Brown-Cohen et al., 2023; Irving et al., 2018), where LLMs are presented with an argument topic and can either argue for the statement or against. We consider a list of standard debate topics (outlines in Appendix C). The action space here is the argument style: namely logos, ethos or pathos, taken from Aristotle's three modes of persuasion that is widely used and studied in debate forums (Cope & Sandys, 2010). Agents in this game have to argue either for or against the given topic, and the game rewards players by scoring the debate with either a 0 or a 1 depending on the validity of the argument as judged by an LLM reward model. We provide examples of debate topics and generations in Appendix C.

### 3.3 Algorithms to Solve Games

Framing dialogue as a game allows us to "solve" the game using variants of existing game-theoretic approaches. More concretely, depending on the actions or strategies we define for the game (e.g., "to be more submissive"), we can use game-theoretic solvers to solve for which strategy is optimal given the current player valuations and game state, and then use the solver output to guide language model generations. For example, a popular solving algorithm is the Policy-Space Response-Oracle (PSRO) method which alternates between two steps. In one step **Policy-Space**, a set of candidate player policies are evaluated in head-to-head matchups against each other and the outcomes (e.g., the rate of winning of each player) are recorded in a normal-form game payoff matrix. The equilibrium of this matrix game is then calculated. In the other step **Response-Oracle**, each player computes a best response to this equilibrium strategy resulting in a new candidate policy, which is then added to the candidate set used in the other step. In the dialogue game, our policies are prompt strings. An *approximate best response* can be generated by sampling new random prompt strings, evaluating them against the current equilibrium, and then returning the one with highest expected payoff. We provide pseudocode for an approximate best response operator in Algorithm 1.

## 4 Evaluations

Given the framework described in the above two sections, we now empirically evaluate the influence of equilibrium solvers on LLMs. Our experiments aim to evaluate to what extent a game-theoretic solver can guide the reasoning process of an LLM. We empirically evaluate these components separately: i.e., the ability of a game-theoretic solver to provide actions for an LLM, and also the ability of an LLM to actually use this input to generate better dialogue. We also qualitatively evaluate the dialogue generations to assess the difference in outputs. We outline each of our evaluations in the section below.

**Do LLMs follow game-theoretic solver outputs?** When an equilibrium solver computes an action that is deemed optimal, this is given as part of the input into the language model's context after which it generates a response that is evaluated as to whether it gets a higher payoff than its opponent. However, aside from evaluating if the LLM wins the game or gets a higher payoff, we first wish to evaluate whether the generations from an LLM do actually reflect the guidance from the solver. For example, when the determined equilibrium action is to "sound submissive", can we evaluate the dialogue strings from the LLM to empirically assess how often they do actually follow this direction to modulate the tone of their generations?

To evaluate this, given an LLM-generated message $m$ conditioned on a prompt formatted with an action $a$, we ask whether $a$ is actually the most likely action conditioned on $m$ using a held-out model $P$, i.e., $a = \arg\max_{z \in \mathcal{A}} P(z|m)$. This held-out model $P$ is a different LLM that is few-shot prompted to perform this task and validated on a test set. We provide details of the model and prompts in Appendix D. Over a 100 samples for each domain, we see that 75% of the time the models do actually follow the guidance of the solver. It is worth noting that the performance of the evaluator-LLM varies based on how much input is given to it i.e., when detailed definitions of how to recognise the tone/actions are given it performs better as tested on the validation set[4]. We pick the best performing evaluator-LLM from this held out set and report results over the LLM-generated messages.

**Are LLMs more strategic under the influence of solvers?** Now that we have confirmed that LLMs do follow equilibrium solver outputs a significant portion of the time, we aim to quantify how often this helps LLM generations. Specifically, we wish to evaluate the improvement of a solver-guided LLM over a baseline LLM that gets no game-theoretic model input but simply the game information as context. We do this by evaluating how often one LLM wins a game i.e., if the action it chooses allows it to get the highest reward. This is determined by obtaining payoffs for the agents at the end of the game, that denotes the outcome dependent on the strategies employed by the players. A higher payoff is desirable, and an optimal strategy for a player is one that allows it to achieve the highest possible payoff in that state of the game.

To evaluate the payoffs at the end of the game, We sample a 100 games in each domain and run our framework to obtain solver-guided LLM generations. For the same games, we also obtain baseline LLM generations i.e., a language model that does receive a strategy/tone it is told to exhibit, but otherwise receives the exact same information about the game in its context. We see that across the three different domains the solver-guided LLMs receive higher payoffs than the baseline LLMs by a 19% margin.

**How good is an LLM as a reward model?** Along with the evaluations above, we discuss the accuracy of the LLM-based reward model. It is difficult to automate the evaluation of our reward model, because it is difficult to extract from the natural language conversation the exact deal (or no deal) that is agreed upon. We provide example positive demonstrations in Appendix D of using in-context learning to allow LLMs to compute rewards in the fruit trading domain.

**Do different game-theoretic solvers provide different improvements?** To assess the influence of different types of game-theoretic solutions, we explore two different algorithmic approaches to approximating equilibria and evaluate the difference in performance under each. We first look at Counterfactual Regret Minimisation (CFR) a popular frameowrk for solving imperfect information games (Zinkevich et al., 2007). In CFR, each player measures how much they could have gained by switching to a different strategy at a given infostate[5] and then attempts to minimize this gap. An important result of regret-minimizing algorithms is that the time average play of players independently running

---

[4]Note there may be cases in which two actions are indistinguishable for a single message (e.g., stern, frank, terse may result in the same message). This is not necessarily a failure of the model, but a natural byproduct of the ambiguity of conversational dialogue. We leave this to future work on better automated evaluators.

[5]i.e., how much they would *regret* not switching *if* they had visited that infostate during a game.

regret-minimizing algorithms converges to a coarse-correlated equilibrium (Gordon et al., 2008). We use OpenSpiel's (Lanctot et al., 2019) CFR solver on the procedurally generated games to obtain the equilibrium actions. This is then added to the prompt of the model to guide its generations. We evaluate the **CFR Gain** metric described above, and also evaluate **Average NashConv** i.e., the pseudo-distance to the Nash equilibriium that measures how much players can gain by deviating from the joint strategy returned by CFR. For both metrics, we see that the LLMs benefit from CFR solver outputs by virtue of not gaining by switching to the baseline LLM strategy. We report results in Table 1 in Appendix J, and see that CFR returns an improved strategy over a baseline LLM.

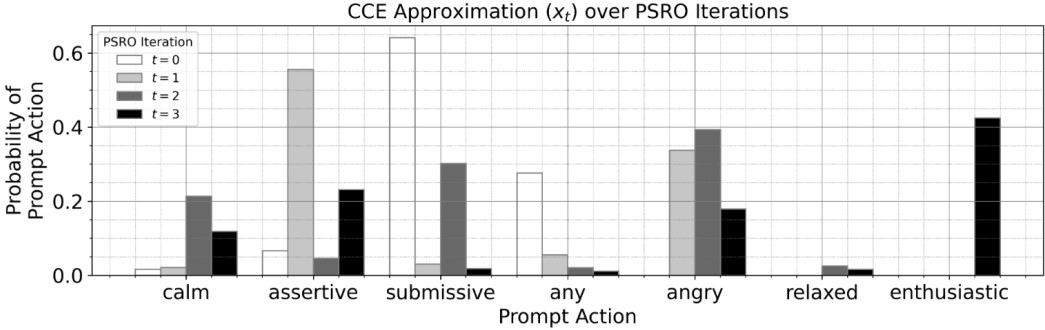

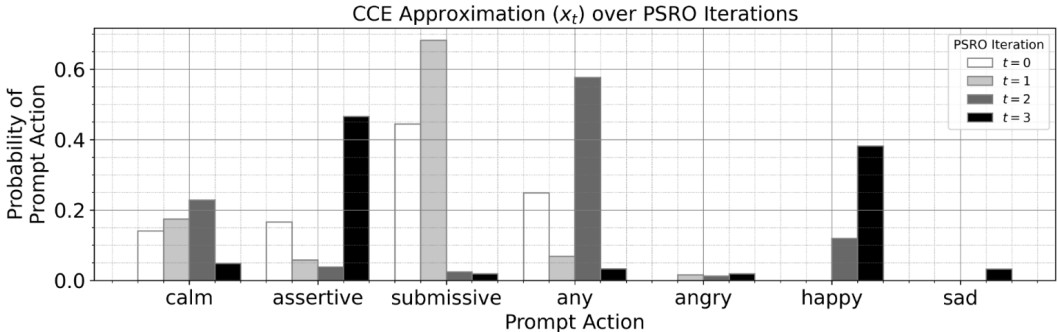

Figure 3: Figure shows performance of PSRO algorithm on the fruit trading domain on the left and meeting scheduling on the right. PSRO begins with the first four candidate prompts ("calm"—"any"). The equilibrium over these prompts is diplayed along with each subsequent equilbrium over the growing candidate set. Recall, each new candidate action was an approximate best response to the previous candidate set (e.g., "angry" was a best response to the equilibrium over "calm"—"any").

We then consider the PSRO algorithm that alternates between solving for an equilibrium of the game and then approximating a best response to this equilibrium. In Figure 3 ($t = 0$), on the left, the fixed-action equilibrium distribution over the initial action set { "calm", "assertive", "submissive", "any" } is reported for the meeting scheduling domain. We solved for this equilibrium using replicator dynamics (Weibull, 1997), an algorithm from evolutionary game-theory with strong connects to regret-minimizing algorithms for normal-form games. The remaining PSRO iterations ($t > 0$) generate approximate best responses ("angry" then "happy" then "sad") along with equilibrium approximations over each new support. This demonstrates the algorithms ability to grow and refine the game's original action space by introducing novel prompt instructions into the LLM's repertoire.

We report the same experiments for the fruit trading game in Figures 3 on the right. For both domains, we see that "submissive" is initially the most probable action at equilibrium—potentially a tactic in which another LLM may exploit it by eliciting a sense of responsibility or fairness (Park et al., 2023). Given the human data LLMs are trained on, this is likely from observed conversations in which one speaker appealed to the other's altruistic tendencies.

Across algorithms, we see that the final NB solution is "calm" whereas "assertive", "angry", and "enthusiastic" (and not "calm") are the predominant actions under the CCE.

**Can the optimal policies constructed by game-theoretic solvers generalize to new domains?**   Lastly, we attempt to see if we can use the game-theoretic solvers to construct optimal target policies that can be learned with imitation learning. If these are possible to learn, this implies a model can be trained to learn optimal behaviour that can generalize to new domains that do not require a solver to explicitly framed over. Therefore, they can serve as the basis for an improvement operator that can work in any new domain.

We consider 200 procedurally generated games, and use 10 iterations of the CFR solver to solve for an equilibrium for each game. We then save vector observations of each information state along with the optimal equilibrium policy returned by CFR for that infostate. This is a length-768 string embedding paired with a distribution over num_tones actions for each game. We call the collection of such pairs our imitation dataset. With this dataset, it is then fairly straightforward to train a neural network policy to predict the equilibrium probabilities conditioned on the observations. We find that a two-layer MLP trained over this dataset is sufficient to learn this distribution (see Appendix E for full architecture and training details of this imitation learning model). This imitation learning model is then compared against an LLM that only plays the action "any" on held out games.[6] We report more detailed results in Appendix E, but we find that more mass lies on our CFR imitation policy under the equilibrium distribution implying it achieves higher payoff than the vanilla policy. Importantly, this implies our proposed approach results in an improved policy.

## Discussion

Our framework suggests that it is possible to use game theoretic models to guide the reasoning of language models in dialogue games. Given the large body of work that finds that LLMs exhibit inconsistent reasoning strategies, we see that a good solution to this problem is to offload the strategic reasoning component to a game-theoretic solver can provide benefits to the LLM in that it generates language that allows it to win against an opponent. Moreover, we show how this framework does not have to be limited to simple games historically studied in the field of game theory. When standard natural language dialogue tasks are framed as a game, we can use existing game-theoretic solvers over such games. The solver outputs can then be used to instruct an LLM how to behave in a way that is strategically optimal. It is worth noting however, that our current approach only guides LLM generations at inference time, rather than incorporating characteristics of game-theoretic reasoning into the training or fine-tuning of LLMs. Future work that incorporates these strategies into the learning process of LLMs (e.g., as part of the objective function of a language model) can further allow more complex reasoning of these models.

However, we also wish to outline the several limitation of this framework. Transitions in the dialogue game we pose are extremely expensive. This is because LLM inference is expensive, both computationally and financially and given the large game trees that need to be created, these calculations can very quickly become an intensive resource-consuming operation. If we want to approximate equilibria or search these game trees efficiently, we will need to push scalable game-theoretic algorithms, e.g., (Burch et al., 2012), to new heights. Additionally, our game-theoretic models are currently limited in several respects: players are assumed to have the same payoffs and action spaces and several assumptions are made on the exact structure of interactions, that is different from many tasks in the real world. Lastly, given the illogical and incorrect reasoning patterns employed by LLMs that are not grounded in the real world (Agnew et al., 2024; Fried et al., 2023) it is up for debate whether their fidelity is sufficient to draw conclusions on real world interactions. Using these models as black-box operators or autoraters for evaluation is often not 100% reliable. If these limitations are addressed, this implies a clear path forward towards guiding language

---

[6]The intention is that an LLM guided to use "any" tone performs similarly to the original LLM with no guidance.

models with "optimal" strategies found by game-theoretic models to pave the way for more intelligent language model agents.

## 5 Related Work

There are two bodies of work most relevant to our focus here. The first is on enhancing the in-context reasoning abilities of LLMs in interactive settings and understanding where their failures lie. The second is on enabling LLMs to reason strategically in complex multi-agent game settings by employing game-theoretic measures for better strategies. We situate our work between these two areas and outline the relevant literature from both sets of work below.

**In-context reasoning capabilities of large language models**   Existing LLMs have been shown to exhibit better reasoning capabilities when this is explicitly taught to them, either at inference time through in-context learning with chain-of-thought or "scratchpad" approaches (Wei et al., 2022; Nye et al., 2021) or at training-time with data containing reasoning traces of humans (Rajani et al., 2019; Shwartz et al., 2020) or even on their own reasoning traces (Zelikman et al., 2022). Additionally, Gandhi et al. (2023) develop an automated "prompt compiler" that constructs demonstrations for LLMs to solve games and Patel & Pavlick (2021) teach models to learn concepts in grounded domains. However, it is worth noting that even methods like chain-of-thought are often unreliable and unfaithful to the generations of models Turpin et al. (2024) and models are susceptible to small variations in inputs that drastically alter behaviour Webson & Pavlick (2021).

**Game-theoretic measures to improve language model capabilities**   There is a substantial line of work focused on allowing LLMs to strategically interact with one another, either in natural language or over a space of actions in a game (Fried et al., 2023). In games that require agents to communicate with one another (e.g., Diplomacy (FAIR et al., 2022)) or even standard games that do not require explicit communication but could benefit from natural language hints (e.g., Minecraft-like games (Fan et al., 2022; Rodriguez-Sanchez et al., 2022)) there have been improvements in performance by using language-guided agents for better reasoning. insights from game-theory have been shown to improve LLM behaviour before. Other aspects of language models have also been shown to improve with insights from game theory. Jacob et al. (2023) use an equilibrium-ranking model to improve the factuality of generated text from an LLM, Patel et al. (2021) show how game-theoretic power indices can improve vocabulary selection for language model training and Ethayarajh & Jurafsky (2021) show how the attention mechanisms in LLMs can be re-understood through Shapley power indices. Complementary to these approaches, our work shows how equilibrium solvers can be used in conjunction with LLMs to improve their reasoning capabilities.

## 6 Conclusion

Our work investigates the extent to which natural language generations from LLMs can be steered towards more strategic behaviour by game-theoretic solvers. This is motivated by the fact that current LLMs do not exhibit coherent and intelligent reasoning strategies, and also that game theoretic algorithms that could solve such problems have so far only had a limited impact on language agents. To address this, we formulate a framework to map natural language dialogue games to a formalism that equilibrium solvers can find solutions over, and show that across three dialogue domains, this improves the LLM generations compared to a baseline LLM that does not have access to game-theoretic models. By doing so in this work, we open the door for a broad body of game theory and multi-agent research—not just on algorithms but also solution concepts and principled strategic reasoning—to pour into the AI guided interactions that pervade humans' daily life. Especially since these conversational models are already becoming a part of users' lives it is imperative that we begin to model and study these interactions to enable large language models to exhibit clear and interpretable reasoning strategies when interacting with human users.

## Ethics Statement

Strategic dialogue agents may be more "rational", but even in well-intended settings, it is known that seemingly benign behavior can lead to poor outcomes for the group, i.e., high *price of anarchy or stability* (Nisan et al., 2007). Moreover, recent work has found that some of the beneficial social norms humans have developed can collapse when people are allowed to leave their coordination decisions to AI assistants (Shirado et al., 2023). In more nefarious cases, strategic agents can exploit others. We should aim to create agents that are both rational, but also generate high welfare. By exploring these interactions in simulation, we can learn how to constrain and regulate agents toward more human-aligned behaviors. Designing algorithms to discover optimal (e.g, max-welfare) equilibria in $n$-player, general-sum games is an active area of research that can help guide the development of more ethical agents.

**Societal Impact**    If the result of this work is that LLMs are, for example, "assertive" $x\%$ of the time and "submissive" $y\%$ of the time, how can we measure and/or predict their effect on human dialogue or sentiment in society at large? When the car was invented, it took decades to mass produce and gain adoption by a significant percentage of the population. LLMs, being a digital technology, could see widespread adoption only a few years after their invention. In contrast to the adoption of cars in the early 20th century, we have the capability to digitally simulate and forecast the impact of large language models. For example, if LLMs are more assertive on average than the human population, will they draw human society towards exhibiting more assertive personalities (Baumann et al., 2020)? If LLMs are more rational and forward thinking, will these strategies similarly influence humans interacting with these systems? It is important to forecast the equilibria of these large techno-societal changes before they happen.

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

# A  Appendix

# B  Background on Extensive-Form Games

Extensive-form games allow us to represent sequential (i.e., temporally-extended) interactions between players. Every game starts with an empty history and players take actions, appending them to the action history, until the end of the game is reached. Formally, an **extensive-form game** is a tuple $\langle \mathcal{N}, c, \mathcal{A}, \mathcal{H}, \mathcal{Z}, u, \tau, \mathcal{S} \rangle$, where $\mathcal{N} = \{1, 2, \cdots, n\}$ is a set of $n$ **players**.[7]. $\mathcal{A}$ is a set of **actions** players can take. $\mathcal{H}$ is a finite set of **histories**, where each history is a sequence of actions (including chance node "actions" or *outcomes*) taken from the start of the game. $\mathcal{Z} \subseteq \mathcal{H}$ is a set of **terminal histories** that each represent a fully played, finished game. A utility or payoff function $u : \mathcal{Z} \to \Delta_u^n \subseteq \Re^n$, where $\Delta_u = [u_{\min}, u_{\max}]$ assigns each player a payoff at the end of the game, and a player identity function $\tau : \mathcal{H} \to \mathcal{N} \cup \{c\}$ indicates whose turn it is. Lastly, there is a set of **states** $\mathcal{S}$ at which players make decisions. For sequential games, $\mathcal{S}$ is a partition of $\mathcal{H}$ since each state contains information of the histories until that point in the game. At each state, there are also legal actions that can be taken that are a subset of all the actions $A(s) \subseteq \mathcal{A}$. Importantly, a history represents the true ground/world state: when agents act, they change this history. For imperfect information games however some actions may be private and not revealed to some players.

---

[7]There is sometimes also a special player, *c*, called **chance** (or "nature")

There are several categories of extensive-form games based on the relationships of payoffs across players: namely **constant-sum**, **zero-sum**, **identical interest** or **general-sum** games[8] Our focus here is on...

## C Dialogue Game Prompts to LLMs

```
%%%%%%%%%%%%%%%%%%%%%%%%%%

Fruit Endowment:
apple: 1
banana: 2
blueberry: 0
kiwi: 0

Fruit Valuations:
apple: 10
banana: 5
blueberry: 1
kiwi: 3

Tone: Use a calm tone.

###########################
Schedule Proposal Message:
from: Bob
to: Suzy
###########################
```

## D LLM-based Reward Model

These examples were generated using the PaLM 2 L (unicorn).

### D.1 No Trade Agreement

Here, we report the final message in a trade dialogue:

```
###########################
Trade Proposal Message:
from: Alina
to: Ciara
###########################

Hi Ciara,

I'm so sorry, but I just don't see a way to make a deal that we can both be
happy with. I really need bananas and blueberries, and I'm not really
interested in kiwis. I know you're disappointed, and I am too. I was really
hoping we could make a trade.

I wish you all the best in finding s...
```

The LLM correctly calculated the rewards / payoffs / utilities as:

```
Utility for player 0 is 0.0
Utility for player 1 is 0.0.
```

---

[8]We provide a detailed overview of each of these categories in Appendix **??**.

## D.2 Trade Agreement

We report the final two messages of a trade dialogue, which suggest a trade agreement has been reached (although ideally the participants would have confirmed this in a final message).

```
############################
Trade Proposal Message:
from: Alina
to: Elroy
############################

Hi Elroy,

Thanks for your response. I'm glad we agree that bananas are more valuable
than kiwis. Unfortunately, I can't trade two apples for one banana. I'm
willing to trade one apple for one banana, or two kiwis for one banana.

Let me know if you're interested in either of those trades.

Thanks,

############################
Trade Proposal Message:
from: Elroy
to: Alina
############################

Hi Alina,

Thanks for your response. I understand that you're not willing to trade
two apples for one banana. I'm willing to accept two kiwis for one banana,
if you're still interested.

Thanks,
Elroy
```

The LLM-based reward model uses chain-of-thought to calculate trade values. We report an intermediate thought in the chain where the LLM has aggregated the information critical to calculating the trade.

```
Alina
fruit_endowment:
apple: 2
banana: 1
blueberry: 1
kiwi: 2
fruit_valuations:
apple: 6
banana: 5
blueberry: 1
kiwi: 1

Elroy
fruit_endowment:
apple: 2
banana: 1
blueberry: 1
kiwi: 2
fruit_valuations:
apple: 6
banana: 5
```

```
blueberry: 1
kiwi: 1

Player Alina: Receives 1 banana Gives 2 kiwis

Player Elroy: Receives 2 kiwis Gives 1 banana
```

From this intermediate information, the LLM then calculates the following utilities:

```
Utility for player 0 is 3.0   # Alina
Utility for player 1 is -3.0  # Elroy
```

### D.3   Incorrect Trade Value

The final two messages of this dialogue imply a trade agreement has not yet been reached and negotiations are ongoing.

```
###########################
Trade Proposal Message:
from: John
to: Elroy
###########################

Hi Elroy,

I would be happy to trade 3 kiwis for 1 apple. Do you think that sounds fair?

Best,
John

###########################
Trade Proposal Message:
from: Elroy
to: John
###########################

Hi John,

Thanks for your offer. I would be willing to trade 1 apple for 4 kiwis.
What do you think?

Elroy
```

Even the LLMs intermediate calculations show no agreement has been reached.

```
John
fruit_endowment:
apple: 0
banana: 0
blueberry: 4
kiwi: 4
fruit_valuations:
apple: 6
banana: 5
blueberry: 1
kiwi: 1

Elroy
fruit_endowment:
apple: 0
banana: 0
```

```
blueberry: 0
kiwi: 6
fruit_valuations:
apple: 6
banana: 9
blueberry: 3
kiwi: 1

No agreement has been reached.

&&&&&&&&&&&&&&&&&&&&&&&&&&&&&&&&&&&&&&&&&&&&&&&&
```

Despite this, the LLM-based reward model hallucinates a trade where Elroy gives John 6 kiwis in return for 4 blueberries. On top of this, the LLM miscalculates the value to Elroy of receiving 4 blueberries; it appears to use John's valuation for blueberries.

```
Elroy receives 4 blueberries, which are worth 1 each. Therefore Elroy receives a
value of 4 in the trade.

Receives: 4

Now we calculate the value of the fruits Elroy gives up in the trade.
Elroy gives up 6 kiwis, which are worth 1 each. Therefore Elroy gives up a
value of 6 in the trade.

Gives: 6

Subtracting the value Elroy gives away from the value Elroy receives, we
find 4 - 6 = -2.

Calculation: Receives - Gives = 4 - 6 = -2.

Value for Elroy: -2.
```

You may also notice that in the final message, Elroy offers to give up an apple, which, according to his endowment, he does not have. This is an issue with the LLM being prompted to generate sensible messages.

## E   Imitation Learning

We used a fully connected neural network with two dense hidden layers of size 256 neurons each and a final dense layer trained against the CFR target probabilities to minimize a cross entropy loss. We trained the policy using $10^4$ steps of Adam (Kingma & Ba, 2014) with a batch size of 128 and learning rate of $10^{-3}$.

## F   Imitation Learning Results

Figure 4a displays the equilibrium distribution in game where a player is given the choice between these two models.

## G   ChatGames

```
1  config = config_dict.ConfigDict()
2
3  num_players = 2
4
5  observations = [
6    obs_utils.Observation(summary.PREFIX, summary.POSTFIX)
```

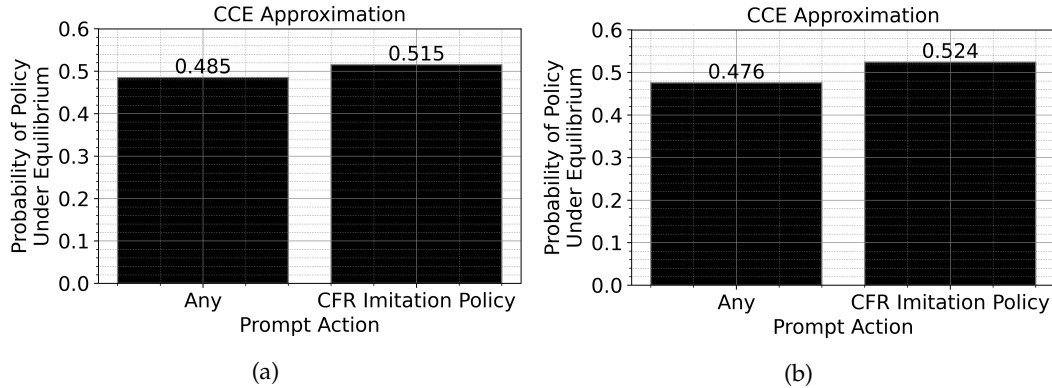

Figure 4: Proof-of-Improvement: Equilibrium Evaluation of Imitation Learned Policy against Baseline LLM in (4a) scheduling a meeting and (4b) trading fruit.

```
7    for _ in range(num_players)
8  ]
```

Next, we define a `header`. The header is a structured object that specifies a string that is to be populated with private information, action information, context, and formatting for prompting the LLM to generate a message.

```
1  header = env_trade_fruit_with_tone_info.HEADER
```

For example, an LLM negotiating fruit might be passed the dialogue history followed by the header below which has already been formatted with its private information (fruit endowment and fruit valuations), its intended action (a "calm" tone), its intended message recipient (Suzy) and its own name (Bob).

```
%%%%%%%%%%%%%%%%%%%%%%%%%%%

Fruit Endowment:
apple: 1
banana: 2
blueberry: 0
kiwi: 0

Fruit Valuations:
apple: 10
banana: 5
blueberry: 1
kiwi: 3

Tone: Use a calm tone.

###########################
Schedule Proposal Message:
from: Bob
to: Suzy
###########################
```

Next, `payoffs` specifies a list of structured payoff objects that can be combined using a user-defined aggregation function. A payoff consists of an LLM prompt, min and max payoff (utility) values, as well as prompts useful for an LLM to transform an input (e.g., dialogue history) into a string containing information more pertinent to payoff calculations.

```
1  payoffs = [payoffs_trade_fruit.PAYOFF]
```

example_names simply consists of a list of names as strings that an LLM uses to generate new names. For example, the names Bob and Suzy could be procedurally generated by an LLM given an initial list of names.

```
1  examples_names = names_trade_fruit.NAMES
```

We can also define the finite set of actions that we would like our game to consider. All players are assumed to have the same set of actions.

```
1  given_prompt_actions = collections.OrderedDict()
2  tones = ["calm", "assertive", "submissive", "any"]
3  given_prompt_actions[header.action_keys[0]] = tones
4  num_tones = len(tones)
```

If we want to procedurally generate new games, we can provide an initial list of examples of private information.

```
1  examples_private_info = collections.OrderedDict()
2  examples_private_info["fruit_endowment"] = [scenario_trade_fruit.ENDOWMENT_A,
3                                              scenario_trade_fruit.ENDOWMENT_B]
4  examples_private_info["fruit_valuations"] = [scenario_trade_fruit.VALUATION_A,
5                                               scenario_trade_fruit.VALUATION_B]
```

Similarly, scenarios are structured objects that can be used to generate more scenarios (i.e., new games). A scenario defines the initial context for a dialogue (e.g., an initial email).

```
1   scenario_a = env_trade_fruit_with_tone_info.Scenario(
2     scenario_trade_fruit.SCENARIO_A,
3     "Bob",
4     "Suzy",
5     scenario_trade_fruit.ENDOWMENT_A,
6     scenario_trade_fruit.VALUATION_A,
7     "calm"
8   )
9
10  scenario_b = env_trade_fruit_with_tone_info.Scenario(
11    scenario_trade_fruit.SCENARIO_B,
12    "Jill",
13    "George",
14    scenario_trade_fruit.ENDOWMENT_B,
15    scenario_trade_fruit.VALUATION_B,
16    "calm"
17  )
18
19  examples_scenarios = [scenario_a, scenario_b]
```

Similar to observations and payoffs, an LLM termination prompt is a structured object that contains prompts for pre-processing the dialogue history using an LLM and then determining whether a given history is terminal by again prompting an LLM.

```
1  llm_termination_prompt = scenario_trade_fruit.LLM_TERMINATION_PROMPT
```

In addition to using an LLM, we can designate terminal histories ($\mathcal{Z}$) by limiting the maximum number of replies per player (below). We can also specify the number of chance node outcomes (LLM seeds). Recall that the action space is combinatorial (recipient $\times$ tone) and so we define the number of actions accordingly. The params dictionary is passed to OpenSpiel to alert it to critical properties of the game that remain fixed (although num_llm_seeds and num_max_replies are specific to our chat_games).

```
1  params = {"num_distinct_actions": num_players * num_tones,
2            "num_llm_seeds": 2,
3            "num_players": num_players,
```

```
4          "min_utility": min([float(p.min) for p in payoffs]),
5          "max_utility": max([float(p.max) for p in payoffs]),
6          "num_max_replies": 1}
7  config.params = params
```

Lastly, we incorporate these definitions into the configuration dictionary. Note that below, we are asking the LLM to generate 10 names (each new game will randomly draw from this list) and 3 of each type of private information (fruit endowment and valuation). The LLM list suffix is a simple modification to help the LLM generate new items given an initial list (e.g., of names).

```
1   config.game = config_dict.ConfigDict()
2   config.game.observations = observations
3   config.game.header = header
4   config.game.payoffs = payoffs
5   config.game.given_prompt_actions = given_prompt_actions
6   config.game.num_names = 10
7   config.game.num_prompt_actions = (num_tones,)
8   config.game.num_private_info = (3, 3)
9   config.game.examples_names = examples_names
10  config.game.examples_private_info = examples_private_info
11  config.game.examples_scenarios = examples_scenarios
12  config.game.llm_list_suffix = "Output: "
13  config.game.llm_termination_prompt = llm_termination_prompt
```

This config dictionary defines the parameters of interaction between players in a dialogue in manner that binds precisely onto a game tree. We use configs like this one later in experiments to study three natural language settings: scheduling a meeting, trading fruit, and debate.

## H   Algorithms and Solution Concepts

---

**Algorithm 1** Shotgun Approximate Best Response

---

**Input:** Focal agent $i$
**Input:** Current joint policy $\pi$
**Input:** Number of shotgun candidates $k$
    $C$ is the current action set with their scores under $\pi$
    **for** $t = 1 \leq k$ **do**
        Prompt LLM to generate new candidate $c_t \cap C = \varnothing$
        Evaluate candidate $c_t$ against policy $\pi_{-i}$ to give score $s_t$
        $C = C \cup \{(c_t, s_t)\}$
    **end for**
**Output:** $c_t$ with max $s_t$

---

Contrast this against the standard PSRO protocol in which an approximate best response is achieved using reinforcement learning (RL) and/or gradient-based optimization. Tasking the LLM with exploring and generating novel candidates avoids these expensive learning procedures while also enabling new, more powerful modes of search, as we explain below.

As an alternative to the random search process just described, we could define a "better response" and build it as a while loop that generates and evaluates a prompt string on each loop iteration. The loop terminates when a better string is found (as measured by an approximate evaluation). See Algorithm 2 for pseudocode.

This rejection sampling approach might be slow. Instead, we could improve the "best response" operator by providing the LLM with information of the action-fitness landscape and ask it to climb it. We provide pseudocode in Algorithm 3. A similar approach was previously explored in Fernando et al. (2023) and Yang et al. (2023).

---

**Algorithm 2** Approximate Better Response

---

**Input:** Focal agent $i$ and its score $s^*$ under $\pi$
**Input:** Current joint policy $\pi$
   **while** $s \leq s^*$ **do**
      Prompt LLM to generate new candidate $c$
      Evaluate candidate $c$ against policy $\pi_{-i}$ to give score $s$
   **end while**
**Output:** $c$

---

**Algorithm 3** Trajectory-Aware Approximate Best Response

---

**Input:** Focal agent $i$
**Input:** Current joint policy $\pi$
**Input:** Number of candidates $k$
   $C$ is the current action set with their scores under $\pi$
   Order $C$ by their scores in ascending order
   Prompt LLM to generate $k$ new candidates in order of ascending score given ranked $C$
   Evaluate new candidates against policy $\pi_{-i}$ to give scores
**Output:** $c_t$ with max $s_t$

---

Lastly, we can blur the lines between game modelling and game solving by prompting an LLM to generate new dimensions of a combinatorial action space. For instance, imagine we constructed an action space consisting only of the tones and styles in which an LLM will generate a response. Is that the full-space in which we would like to explore writing a successful message?

---

**Algorithm 4** Categorical Approximate Best Response

---

**Input:** Focal agent $i$
**Input:** Current joint policy $\pi$
**Input:** Number of candidates per category $k$
**Input:** Number of category candidates $k'$
   $C$ is the current set of action categories with their (Nash) average scores under $\pi$
   Order $C$ by their scores in ascending order
   Prompt LLM to generate $k'$ new candidate categories in order of ascending score given ranked $C$
   Prompt LLM to generate $k$ candidates for each new action category
   Evaluate new candidates against policy $\pi_{-i}$ to give scores
**Output:** Category with highest average score

---

# I   Tractability of Solution Concepts

Solution concepts have different existence and computability properties depending on the properties of the game model class. In particular, our later choice of a discrete action set over a continuous one, e.g., the set of weights of the underlying LLMs, is inspired by recent computational hardness results for computing approximate local Nash equilibria in the case of continuous strategy sets even in two player zero-sum games. Specifically, Daskalakis et al. (2021) show that, in the case of constrained min-max optimization problems with nonconvex-nonconcave objectives and linear constraints, even when the objective is a Lipschitz and smooth differentiable function, deciding whether an (approximate) min-max point exists, is NP-hard. Even when an approximate local min-max point of large enough approximation is guaranteed to exist, finding one such point is PPAD-complete.

In follow-up work, Assos et al. (2023) show that PSRO-style (Lanctot et al., 2017) approaches actually lead to tractable notions of approximate local Nash equilibria even in continuous games. PSRO (Policy-Space Response-Oracle) algorithms start with a set of initial policies

---

**Algorithm 5** Prompt-Space Response-Oracles

---

**Input:** $C$ where $C_i$ is the initial prompt action set (singleton) for player $i$
**Input:** $h$ containing hyperparameters for approximate best response operator BR
  Compute expected payoff tensor $P$ over joint action(s) $C$
  $\pi$ is uniform meta-strategy profile over $C$
  incomplete = True
  **while** incomplete **do**
    **for** player $i \in [N]$ **do**
      $c_i \leftarrow \text{BR}(i, \pi, h)$, e.g., Algorithms (1-4)
    **end for**
    **if** $c_i \in C_i \ \forall i \in [N]$ **then**
      incomplete = False
    **else**
      $C_i \leftarrow C_i \cup c_i \ \forall i \in [N]$
      Compute expected payoff tensor $P$ over joint actions $C$
      $\pi \leftarrow$ meta-strategy w.r.t. $P$
    **end if**
  **end while**
**Output:** $(\pi, C, P)$

---

for each player and then alternate between two steps. In one step, a normal-form *meta*-game (e.g., matrix game) is constructed in which each player may select a policy to play the game on their behalf. A *meta*-solver (e.g., Nash solver) returns a solution (e.g., Nash equilibrium) of this *meta*-game. In the next step, each player computes an approximate best response to this *meta*-strategy, meaning each player attempts to improve their payoff in the game assuming the other players are fixed to sampling policies according to the *meta*-strategy to play on their behalf. The process repeats until no player is able to gain by deviating. This route of defining a discrete mesh over a continuous strategy space is something we exploit later in Section 3.3 to construct algorithms that act directly in "prompt-space".

## J   CFR Solver Results

|  | # of Samples | Min / Max Payoff | NashConv | CFR Gain |
|---|---|---|---|---|
| **Debate** | 328 | 0/1 | 0.024 | 0.106 |
| **Schedule Meeting** | 67 | 0/20 | 0.417 | 1.596 |

Table 1: Average NashConv and CFR gain for debate and meeting scheduling domains. NashConv is a pseudo-distance to Nash equilibrium and measures how much players can gain by deviating from the joint strategy returned by CFR. CFR gain measures how much a player can gain by switching to the CFR strategy from the baseline LLM strategy.

