# OpenReview forum: "Steering Language Models with Game-Theoretic Solvers"
_ICML.cc/2024/Workshop/Agentic_Markets — Agentic Markets @ ICML'24 Poster_

### Official Review · Reviewer_gdZp · 2024-06-12
**This paper explores the ability of Large Language Models (LLMs) to generate language when guided by Game-Theoretic equilibrium solvers. The focal point of this research paper is to analyse and assess whether LLMs provide more strategic response and less exploitable dialogue generations, when guided by equilibrium solvers.**

**Rating:** 6
**Confidence:** 3

**Review:**

**Research Paper Summary**:

·       The paper discusses that even though existing LLMs have achieved great performance on natural language tasks, there are failures in the reasoning capabilities of these models.

·       It cites multiple previous studies that showcase that despite LLM-generated text appearing credible and reasonable, the reasoning strategies used are not always rational and consistent.

·       The author(s) attempt to establish through evaluation that “LLMs that do follow game-theory solvers result in dialogue generations which are less exploitable than the control (no guidance from solvers)” in the three negotiation domains used.

·       These three negotiation domains are – 1. scheduling meetings, 2. trading fruit and 2. Public debate.

·       The evaluation is done in two areas – 1. whether the dialogue generations from the LLM reflect the guidance from the equilibrium solver and 2. whether the LLM generates better and more strategic dialogue using the input from the equilibrium solver.


**Quality**:

 ·  The paper is well-written and easy to follow.

 ·  The paper proposes an interesting idea.

**Clarity**:

·  This paper is structured in a clear and coherent way that follows a natural order of progression and uses concise language which effectively conveys the central arguments.

**Originality**:

·  Not much extensive work has been done yet on the impact of game theoretic solvers’ input into LLMs and how that affects or enhances their performance in language generation. With increased research and adequate resources, this domain could possibly yield significant results.


**Significance**:

·  It is important to not treat LLMs as black boxes and only be interested in the input and output of the LLM, it is important to assess the reasoning strategies that guide those outputs and address those shortcomings, for longer term better performance and longevity of the use of LLMs (with human welfare in mind). The paper attempts to do that using game theoretic solver’s input.

·  The “Ethics Statement” and “Societal Impact” sections of this paper resonated well with me.


**Pros**:

1.  The problem that this work attempts to address is very relevant since currently the guardrails for LLM outputs can be easily manipulated. If the reasoning strategies of LLMs can be understood better and made to be more consistent and less exploitable, that would be beneficial for overall longevity and use of LLMs.

2.  Paper is well written. Idea is clearly explained in concise language, and follows a natural order of progression, making it easy on the reader.


**Cons**:

1.  There are some assumptions made in the game-theoretic model (for example players have the same payoffs). Real world tasks given to the LLM could be different and the assumptions might not hold, the author(s) don’t explain how to address these shortcomings.

2.  The results derived are not clearly explained.


**Questions/Comments**:

1.  Page 6, Paragraph 4 of this paper (“Evaluations” section) mentions that “across the three different domains the solver guided LLMs receive higher payoffs than the baseline LLMs by a 19% margin”. It is not explained how the author(s) came up with the 19% margin.

2. Figure 3 is hard to understand. The graph is confusing with no clear demarcations between PSRO algorithm performance on the fruit trading domain and meeting scheduling domain. Please clarify which one is for fruit trading and which one is for meeting scheduling.

3. How is the LLM reward model impacting the evaluation/results?

4. The authors mention that the prompt actions were evaluated. It isn’t clear how that was done. Was it done manually? If so, don't the evaluators' biases creep into the process?

5. A finite space was used for the set of possible actions. How is the space selected? More details would be appreciated.

6. The code generation is based on prompts. How are the prompts engineered? Were alternate prompts conveying the same message tested? If so, what were the actions selected?

---

### Official Review · Reviewer_DfxT · 2024-06-14
**First step towards providing LLMs with game theoretic equilibrium solvers to yield higher payoffs**

**Rating:** 7
**Confidence:** 4

**Review:**

## Summary
This paper addresses the limitations of current SOTA LLMs in rational, game-theoretic based reasoning strategies. LLMs, when combined with game-theoretic equilibrium solvers, are able to outperform unguided LLMs to achieve higher payoffs in 3 settings developed: meetings, trading, and debate.

## Strengths
- This represents introductory work in an important direction: enabling LLMs to reason strategically. Additional work will build on top of this and possibly yield better training or scaffolding methods that will make the approach of using guidance from equilibrium solvers scalable to more settings.
- Dialogue tasks are framed as extensive-form games and the environments are a useful first step, especially supporting the empirical results in the paper.

## Weaknesses
- Limitations related to scalability of this method (using equilibrium solvers) for other environments and action spaces that are so restricted will not appear in the real world. Limited real world applicability of the current setting, although this is outweighed by the paper presenting a first step towards combining a strategic problem solving approach and language modeling tasks.
- Limited novelty with respect to the technical solutions offered. Combining language models with tools generally improves performance on narrow, well-defined tasks.